# The Sardinian Bitter Honey: From Ancient Healing Use to Recent Findings

**DOI:** 10.3390/antiox10040506

**Published:** 2021-03-24

**Authors:** Ignazio Floris, Michelina Pusceddu, Alberto Satta

**Affiliations:** Department of Agricultural Sciences, University of Sassari, Viale Italia 39A, 07100 Sassari, Italy; mpusceddu@uniss.it (M.P.); albsatta@uniss.it (A.S.)

**Keywords:** strawberry tree honey, origin, characteristics, traditional uses, antioxidant properties

## Abstract

Sardinian bitter honey, obtained from the autumnal flowering of the strawberry tree (*Arbutus unedo* L.), has an old fame and tradition in popular use, especially as a medicine. Its knowledge dates back over 2000 years, starting from the Greeks and Romans to the present day. There are many literary references from illustrious personalities of the past such as Cicero, Horace, Virgil, and Dioscorides, until recent times, associated with the peculiar anomaly of its taste, which lends itself to literary and poetic metaphors. The curiosity of its bitter taste is also what led to the first studies starting in the late 1800s, aimed to reveal its origin. Other studies on its botanical source and characteristics have been carried out over time, up to the most recent investigations, which have confirmed its potential for use in the medical field, thanks to its antioxidant, antiradical, and cancer-preventing properties. These benefits have been associated with its phenolic component and in particular with the prevailing phenolic acid (homogentisic acid). Later, other strawberry tree honeys from the Mediterranean area have also shown the same properties. However, Sardinian bitter honey maintains its geographical and historical identity, which is recognized by other Mediterranean cultures.

## 1. Introduction

The bitter honey typically produced in Sardinia has aroused the interest and curiosity of many learned and distinguished people, who have expressed different, often conflicting, opinions on the cause of its bitterness as well as on its quality and therapeutic properties [1,2]. Among Romans, Cicero (Marcus Tullius Cicero, 106–43 B.C.) mentioned the Sardinian bitter honey by saying: “*Omnia que Sardinia fert, homines et res, mala sunt. Etiam mel, quod in ea insula abundat, amarum est*” (Sardinia was a bad island, everything in the island was ugly and even its abundant honey was bitter). Horace (Quintus Horatius Flaccus, 65-8 B.C.) in a passage of his *Ars poetica* associated this honey with various unpleasant things like an out-of-tune concert and a fat ointment: “*Ut gratas inter mensas symphonia discors, Et crassum unguentum, et sardo cum melle papaver Offendunt*”. In the Bucoliche (Ecloga VII), Virgilio (Publius Vergilius Maro, 70-19 B.C.) countered the sweet thyme honey of the Iblei (Mounts of Sicily) to the Sardinian bitter honey to express the opposite feelings between Coridone and Tirsi: “*Nerine Galatea, thymo mihi dulsior Hyblae*...” and “*Immo ego sardois videartibi amarior herbis*...”. With regard to the properties of this honey, Dioscoride (Pedanius Dioscorides, A.D. 1st century), a physician, pharmacologist, and botanist, in his treatise *De Materia Medica*, enhanced the medicinal qualities of the Sardinian bitter honey, especially regarding the skin: “*Probatissimum Sardiniae mel amarum apes ex Absinthio depromere et with cere/conficere dudum tradidit*” (Dioscorides lib. II cap. CII.). Concerning Sardinian humanistic literature, the most significative example is probably the novel of the writer Salvatore Cambosu (1895–1962), *Miele amaro*, representing a composite historical, ethnological, and poetic collection on Sardinia.

About its origin, many hypotheses have been made since ancient times. It has been attributed to various sources: strawberry tree (*Arbutus unedo* L.), tree wormwood (*Artemisia arborescens* L.), flax-leaved daphne (*Daphne gnidium* L.), and rue (*Ruta graveolens* L.). The first reference on its bitterness was by Perroncito [3], at that time a researcher and professor at the University of Sassari. Assuming that the bitterness came from some special fermentation similar to that of glucose in grapes during winemaking, he carried out microbiological and chemical investigations that showed large quantities of a saccharomycete but no trace of alcohol. He therefore concluded that the bitterness did not come from alcoholic fermentation, but might have come from bitter ingredients in tree wormwood flowers. About 40 years later, another chemist from the University of Sassari investigated the possible origin of this honey’s bitterness [4]. In this case, presuming the origin of the strawberry tree and starting from Xenophon’s assertions in the Anabasis that the nectars of certain flowers (rhododendron, jasmine, hemlock, oleander, and aconite) transfer their properties to honey, he considered that the bitter taste was due to some substance contained in the plant from which the honey originated. On this basis, he proved that arbutin glycoside was present in honey and strawberry tree (flowers and leaves), accompanied by small amounts of methyl-arbutin. He assigned the possible origin of bitterness to arbutin, not without reservations: “Even if arbutin is found in honey and strawberry trees, it is not certain that the bitterness comes from this substance alone”. Recently, Arbutin was quantified in 83% of samples, so it might be a marker for strawberry-tree honeys from southern Europe [5]. The same study has highlighted other chemical characteristics and properties of strawberry tree honey, all useful for its authentication, differentiation, and valorization for possible future health uses, not only as functional food, but also for cosmetics and pharmaceuticals thanks to its biological properties.

## 2. Profile and Features

The strawberry tree (*Arbutus undo* L.) (Figure 1) is the main source of the renowned bitter honey, of which Sardinia is probably the region with the most significant production in the world [6,7].

Other Italian regions of production of this honey are Liguria, Tuscany, and Campania. In the Mediterranean basin, it is produced also in Corsica, Portugal, Spain, Albania, Croatia, and Turkey.

Strawberry tree is an evergreen species, usually a shrub, that grows spontaneously mainly around the Mediterranean basin. In the temperate area of Europe it occurs along the west coast, reaching its most septentrional limit in the northwest of Ireland. Moreover, its distribution spans in north-eastern Africa and western Asia. Despite the seasonal limitations of the flowering period (October–January), the considerable spread of the shrub plant association (maquis shrubland) throughout the island constitutes a considerable honey yield potential. The melliferous potential and the nectar secretion trend of the strawberry tree has been estimated in Sardinia [8]. According to this study, the honey yield potential referred to the maquis shrubland is 40 kg/ha (attributable to the second productivity class), and the production period is optimal from the first decade of November to the second decade of December (Figure 2), but the flowering is extended to a much longer period. Moreover, its economic importance has increased considerably from its market prices (four to eight times higher than those of other honeys).

From the sensory point of view [10], bitter honey is amber in liquid form, while it takes on a hazelnut to light brown color when crystallized. Crystallization is rapid and often irregular due to excessive humidity. The smell is medium to very intense, pungent, and characteristic of burnt leather, bitter herbs, and especially coffee. The flavor is not very sweet initially and then decidedly bitter, normally or decidedly acidic, and astringent. The aroma is very characteristic, similar to rhubarb, and very persistent, especially in the bitter component. In crystallized honey, the consistency is creamy with fine and very soluble crystals.

Based on physicochemical data, bitter honey shows generally a low diastatic activity (6.22 ± 1.26 DU) and high levels of moisture content (19.79 ± 1.87%) and acidity (29.59 ± 3.85 meq/kg), in addition to high contents of gluconic acid (10.44 ± 2.54 g/kg) and phenols (460 ± 84 mg/kg) [9,11,12,13].

Examination of the sediment of honeys produced in Sardinia (Figure 3) has made it possible to identify a total of 65 different types of pollen [14], among these, *Arbutus* is underrepresented pollen. In addition to typically autumnal or winter flowering species (*Smilax*, *Hedera*, *Asparagus*, *Inula*, *Scilla*, *Rosmarinus*, *Rhamnus*) are associated pollens from spring (e.g., *Cistus*) or summer (e.g., *Eucalyptus*) flowering, due to secondary or tertiary contamination. The most frequent pollens are represented in Figure 4.

Quantitatively, these honeys usually fall into class I with an average number of granules in 10 g equal to 10,400 ± 5700 [15] and with percentages of *Arbutus* pollen varying from 7.3% to 81.6%. However, in some cases, due to the frequent incidence of over-represented pollens such as *Eucalyptus*, the absolute number of pollen granules can vary considerably, even reaching the III class of representativity (PK/10 g >100,000).

## 3. Floral Markers

Many investigations were performed with the aim to evaluate the possibility of characterizing botanical or geographical origin of honey by using specific chemical markers, e.g., on the base of the analysis of the volatile compounds, phenolic acids, flavonoids, carbohydrates, amino acids, and other constituents. According to literature, in the case of strawberry tree honey, an important role could be played by phenolic compounds [16,17,18]. The phenolic compounds are secondary metabolites of the plants where they play the important role of protection against stress oxidative and related damages. The main structural feature responsible for the antioxidant and free radical scavenging activity in this case are the phenolic hydroxyl groups, which are able to donate the hydrogen atoms as well as electrons to arrest the production of free radicals, thus stopping the oxidative process.

Via the nectar, collected by foraging bees, these substances are transferred in the hive, where they are maintained and concentrated in the elaboration process of honey. The qualitative and quantitative differences in the phenolic profile of honeys from diverse floral sources depend on the natural variability of these compounds between plant species and their geographical distribution, making it possible to associate one or more of these compounds with the botanical [16,17,19] or geographical origin of the honey [20,21].

The presence of phenolic acids contributes to the functional and therapeutic properties of honey. Various phenol acids and their derivates were detected in honey in relation to its botanical origin [22]. Gallic acid is one of the most frequent in honey [23].

Recent literature reports numerous attempts to provide the health-promoting properties of monofloral honeys associated to phenolic compounds and to their floral or geographical origin. The potential role of honey phenolic compounds in treating cardiovascular diseases in humans was also described [24]. Diverse phenolic and flavonoid compounds, including quercetin, kaempferol, apigenin, and caffeic acid, hence improve cardiovascular diseases through various mechanisms, such as by decreasing oxidative stress and inhibiting blood platelet activation. Moreover, the content of flavonoids and phenolic acids in honey plays a key role in antimicrobial capacity, anticancer, and antidiabetic activities, and against asthma, and a beneficial effect was also demonstrated on the gastrointestinal system [25].

The Sardinian bitter honey from strawberry tree is rich in phenolic compounds [11,26]. In strawberry tree nectar and corresponding honey, the homogentisic acid (2,5-dihydroxyphenylacetic acid) (HGA) is the most abundant and was detected as a possible chemical marker of strawberry tree honey. The amount of this phenolic acid in Sardinian bitter honey was estimated from 197 to 540 mg/kg (378 ± 92 mg/kg) in relation to the melissopalynological characteristics of the honey samples [16]. The same phenolic acid in the strawberry tree nectar was detected in the amount of 165 mg/kg, proving that this acid in honey came from nectar. Since the average water content of the bitter honey was 18.9 ± 1.9% and that of the nectar was 64%, the homogentisic acid should have increased by a factor of 2.25 (from 165 to 371 mg/kg) while transforming the nectar into honey. Another study confirmed that HGA is the most abundant phenolic compound in this honey, with an average amount of 414.1 ± 69.8 mg/kg [26].

HPLCDAD-MS/MS analysis of strawberry tree honey samples, preselected by sensory evaluation and melissopalynological analysis, showed that, in addition to the above-mentioned acid, there were other high levels of substances useful for the botanical classification of this monofloral honey. Two of these compounds were isolated and identified as (±)-2-cis,4-trans-abscisic acid (c,tABA) and (±)-2-trans,4-trans-abscisic acid (t,t-ABA). A third compound, a new natural product named unedone, was characterized as an epoxidic derivative of the above-mentioned acids. Structures of c,t-ABA, t,t-ABA, and unedone were elucidated on the basis of extensive 1D and 2D NMR experiments, as well as HPLC-MS/MS and Q-TOF analysis (Figure 5).

In the above honeys, the average amounts of c,t-ABA, t,t-ABA, and unedone were 176.2 ± 25.4, 162.3 ± 21.1, and 32.9 ± 7.1 mg/kg, respectively. Analysis of the *A. unedo* nectar confirmed the floral origin of these compounds found in the honey [26]. Abscisic acid is known to be present in other floral nectars and honeys [27]. This compound was found in heather honey (t,t ABA: 5.58 ± 2.61 mg/kg; c,t ABA: 5.71 ± 2.43 mg/kg); in the case of Portuguese heather honey, the amounts of abscisic acid isomers ranged from 25.00–166.00 mg/kg [28]. It was also present in smaller amounts in rapeseed honey (t,t ABA: 0.43 ± 0.34 mg/kg; c,t ABA: 0.56 ± 0.43 mg/kg), lime-tree honey (c,t ABA: 2.10 ± 1.82 mg/kg), and acacia honey (t,t ABA: 0.18 ± 0.18 mg/kg; c,t ABA: 1.43± 0.42 mg/kg). Probably, apart from the plant species, the environmental conditions where the plants grow can affect the presence in nectar and therefore the content of these compounds in honey.

The amounts of the above four markers of the floral origin of strawberry tree honey were measured in the nectar of the strawberry tree throughout the flowering periods of two years (Figure 6) and at various sampling sites.

The data obtained show that, despite the existence of a good correlation level between all the markers found, only homogentisic acid appears to be the most reliable marker of botanical origin of strawberry tree honey thanks to its higher concentration with respect to those measured for isomers of abscisic acid and unedone [18].

## 4. Antibacterial Activity

The antibacterial activity of honey has been well known since the ancient times [25], and is expressed on both Gram-positive and Gram-negative bacteria, although the first are more sensitive. The disk-diffusion test and the evaluation of the minimum inhibitory concentration (MIC) against diverse bacterial agents was commonly used. In general, monofloral honeys showed greater antibacterial effect than blossom ones [29]. This may be due to some specific phytochemicals that came from the plant source, associated to phenolic or other compounds [30].

The honey antibacterial activity is classified as peroxide and non-peroxide components [31]. Concerning non-peroxide activity, some non-specific physical chemical properties of honey, such as low water activity, osmotic effect, acidity, low pH, and low protein content, contribute to preventing bacterial growth [32,33]. About the peroxide component, glucose oxidase represents the main carbohydrate-metabolizing enzyme mainly added to nectar by bees during elaborating honey, which is responsible of the conversion of glucose into hydrogen peroxide (H_2_O_2_) and gluconic acid under aerobic conditions [34,35]. Factors that affect H_2_O_2_ accumulation are inactivation of the enzyme glucose oxidase by exposure to heat, light [36,37], or catalase [38,39]. Regardless, peroxide component is also present in nectar [40], and substantial variation in its accumulation among different nectar samples was observed [41,42]. It is then possible that nectar-derived peroxidase influences this activity in different honeys.

In the case of Sardinian bitter honey, both the high content of gluconic acid, probably as a result of the action of the enzyme glucose oxidase on glucose with the production of H_2_O_2_, and the high content of polyphenols can explain the natural antimicrobial activity of this honey. Another factor, which would favor the action of glucose oxidase, could be the higher humidity of nectar and honey due to the autumnal period of production, considering the good relationship observed between the antibacterial activity of diluted honey samples and the level of H_2_O_2_ that accumulated in them [43].

In a first investigation [44], the antibacterial activity of 15 samples of bitter honey produced in Sardinia were *in vitro* evaluated against the bacterial strains gram+ *Staphylococcus aureus*-ATCC^®^ 25923™ and gram- *Escherichia coli*-ATCC^®^ 25922™, *Pseudomonas aeruginosa*-ATCC 27853, and *Klebsiella pneumoniae*-ATCC 13853 (Culti-Loops^®^), according to the suitably modified method by Stompfay-Stitz and Kominos [44,45]. The results, expressed as growth levels of the four bacterial strains over 24 h and minimum inhibitory concentrations (%) showed *E. coli* as the most resistant strain, followed by *P. aeruginosa*, *S. aureus*, and *K. pneumoniae*. The best five botanically characterized honey samples determined growth levels, in terms of colony-forming unit according Lindner [46], of 40.3 ± 3.9, 31.9 ± 4.9, 4.3 ± 2.1, and 4.2 ± 1.7 for *S. aureus*, *E. coli*, *P. aeruginosa*, and *K. pneumonie*, respectively (Figure 7).

The same honey samples showed minimum inhibitory concentration, expressed as percentages (*v/v*), of 13.3 ± 2.6%, 4.2 ± 1.7, 3.8 ± 0.7%, and 3.8 ± 0.7% for *P. aeruginosa*, *E. coli*, *K. pneumonia*, and *S. aureus*, respectively (Figure 8). Literature sources indicate that the antibacterial activity of honey is highly dependent on the floral source [48]. It is interesting to note that the degree of inhibition of bitter honey is significantly higher against *K. pneumonia* and *S. aureus* than against *E. coli* and *P. aeruginosa* with inhibitory activity at very low concentrations compared to what has already been reported against the same species of bacteria [31,43,49].

In another investigation [50], carried out on 90 Sardinian honey samples, six of which were from strawberry trees, against *Bacillus subtilis* and *Bacillus stearothermophilus* var. *calidolactis*, the results showed that, on average, honeys had a greater inhibitory action against *B. stearothermophilus* (34.4%) than *B. subtilis* (2.2%). Furthermore, the inhibitory action was reduced after heat treatment and addition of catalase. However, in the case of strawberry tree honeys, it always remained unchanged even after heat treatment at 80 °C. In this case, the antibacterial activity is probably due entirely to non-peroxide components [48]. The findings of the melissopalynological analyses confirmed the close relationship between the inhibitory action and the botanical origin of the honey.

The relation between botanical or geographical origins and antibacterial activity has been extensively documented in literature [33]. Most references reports that antibacterial activity of honey depends considerably on the floral source, particularly concerning monofloral honeys and in relation to their chemical composition.

Based on the most recent literature on this topic [43,49], the Sardinian bitter honey would appear to have an average antibacterial activity comparable to or greater than other honeys tested, including manuka honey. However, a comparison between honeys, although very useful, is very difficult, because the data are not always directly comparable, as they relate to different bacterial strains and to diverse methods of measurements of antibacterial activity [43]. Consequently, the results will depend largely on the technique that needs to be considered when comparing different methods. For example, in relation to a comparable measure such as the minimum inhibitory concentration, the values recorded for manuka honey and other honeys against isolates of *P. aeruginosa*, ranging from 10–20% (*v*/*v*) corresponding more or less to that of Sardinian bitter honeys (on average 17.4%), while concerning various strains of *S. aureus*, the Sardinian bitter honey showed MICs below 10% (*v/v*) (on average 8.2%) compared to average values ranging from 10–30% (*v/v*) of other honeys, including manuka honey [51]. Concerning other methods where honey is incorporated into the nutrient agar or into the nutrient broth or a solution of honey is applied to the center of the agar plate previously inoculated with a microbial culture, in order to estimate the bacterial sensitivity to the honey, results are not always directly comparable and must therefore be properly judged and analyzed. For instance, in the case of the antibacterial activity expressed as diameter of inhibition by the plate diffusion technique, the Gram-positive bacterium *S. aureus* was found to be the most sensitive with an average diameter of inhibition of the honey samples from different floral origins of 32.6 mm, while against Gram-negative bacteria as *E. coli*, *P. aeruginosa*, and *K. pneumoniae*, the sensitivity to different honey samples was on average of 24.3, 23.8, and 23.5 mm, respectively [48]. Regarding Sardinian bitter honeys, the evaluation was made incorporating the honey into the nutrient agar and then evaluating the growth levels of the same bacterial species by visual assessment on the agar plate [46], proving also in this case a highest effect against the Gram-positive bacterium *S. aureus*, and versus *K. pneumoniae* among the Gram-negative bacteria tested (Figure 7).

## 5. Antioxidant Activity and Other Health Benefits

Oxidative stress on key biomolecules can lead to metabolic disorders and thus promote the development of many diseases and dysfunctions. The body’s antioxidant defenses depend not only on pathological situations, but also on certain physiological conditions. The intake of antioxidant compounds through the diet can counteract the effect of oxidizing molecules such as free radicals, reducing oxidative stress [25]. From a medical point of view, Fe^3+^ plays important roles in several harmful oxidation processes within the human organism, then reduction of Fe^3+^ to Fe^2+^ can reverse this process. In order to determine the reduction ability of Fe^3+^ to Fe^2+^, the so called “ferric-reducing antioxidant power” method (FRAP), a colorimetric method, is widely used [52]. The color changes are spectrophotometrically measured at 593 nm, and the results expressed as FRAP units, which are defined as the reduction of 1 M ferric ion to one ferrous ion.

The antioxidant effect is a well-studied bioactivity for natural-compound and can be determined by numerous methods based on scavenging free radicals (e.g., hydroxyl, superoxide), performing typical reduction reactions (e.g., reduction Fe^3+^ to Fe^2+^), or inhibiting pro-oxidant enzymes [53,54].

Honey, as a functional food, was shown to contribute to the reduction of oxidative stress thanks to the presence of antioxidant molecules in its composition, including phenolic compounds. The antioxidant activity of honey depends on various components, both enzymatic (catalase, glucose oxidase, peroxidase) and non-enzymatic (ascorbic acid, a-tocopherol, carotenoids, amino acids, proteins, Maillard reaction products, flavonoids, and phenolic acids) [47,48,49]. The amount and type of these components depend on the botanical origin of the honey [25] and a close relationship between the honey antioxidant activity and total phenolic content has been proven [55,56,57]. Phenolic compounds stabilize free radicals with hydrogen from the hydroxyl groups; hence, their antioxidant action depends on the number of hydroxyl groups [58].

In the Sardinian bitter honey, there are several substances that may be involved in this type of biological activity, such as phenolic compounds, particularly the homogentisic acid (2,5-dihydroxyphenylacetic acid) (HGA) [16], which has shown antioxidant and antiradical activities as well as a protective effect against thermal degradation of cholesterol. The antioxidant (FRAP test) activity of HGA was compared with those of other phenolic compounds like gallic acid and resveratrol and two typical antioxidant additives, vitamin C (ascorbic acid) and butylated hydroxyanisole, resulting in 15.6 ± 0.4 mmol Fe^2+^/g. This value is higher than those showed by resveratrol, the well-known molecule found in grapes and red wine, and comparable to those of ascorbic acid and BHA, typical food preservatives [58].

For the strawberry tree honey, the antioxidant FRAP test gave a result of 11.7 ± 1.7 mmol Fe^2+^/kg (corresponding to 11,700 ± 1700 µmol Fe^2+^/kg), the highest compared with honeydew, heather, eucalyptus, asphodel, and citrus spp. honeys [58]. Moreover, good correlation coefficients were found between total phenol amount/FRAP activity (r = 0.9636) and total phenol amount/DPPH activity (r = 0.8920) [58].

The above average value obtained for the strawberry tree honey (11,700 ± 1700 µmol Fe^2+^/kg) is much higher (approximately from three to 18 times) than that found in other honeys from Poland (Rape, Tilia, Goldenrod, Dandelion, Buckwheat, Multifloral, Nectar-Honeydew, Coniferous honeydew, and Leafy honeydew) [59], expressed as µmol TE/kg, but roughly comparable [60]. Strawberry tree honey expresses much higher values also in comparison to various Thai honeys (605.55–4016.67 µmol Fe^2+^/kg) and the manuka honey (4342.59 µmol Fe^2+^/kg) [61].

To better assess the antioxidant activity, strawberry tree honey was also tested for its protective effect in a biochemical assay of oxidative stress, namely the thermal (140 °C) solvent-free degradation of cholesterol at 1 and 2 hs. Antioxidant activity was reported as percentage of cholesterol protection, calculated considering the percent of sterol consumption in the presence of the antioxidant with respect to total cholesterol consumption without antioxidant. The strawberry tree honey showed a significant cholesterol degradation inhibitory activity, with 30% of protection (25 µg and at the 1 h) and 55% of protection (50 µg and 2 hs) [58].

The pre-treatment with homogentisic acid significantly preserved liposomes and low-density lipoproteins from Cu 2+-induced oxidative damage at 37 °C for 2 h, inhibiting the reduction of polyunsaturated fatty acids and cholesterol, and the increase of their oxidative products. This phenolic acid had no toxic effect in human intestinal epithelial Caco-2 cells within the concentration range tested (5–1000 μM). In brief, HGA is a natural antioxidant, present in relative high amount in Sardinian bitter honey, becoming able to produce a significant protective effect comparable to those of other known antioxidants [58].

More recently, Afrin et al. [51] proved for Sardinian bitter honey a higher antioxidant capacity and cytotoxic properties against human colon adenocarcinoma and metastatic cell lines in comparison with manuka (*Leptospermum scoparium*) honey. Both these honeys induced cytotoxicity and cell death in a dose- and time-dependent manner against the above cells, with less toxicity on non-cancer cells. Compared to manuka honey, bitter honey showed more effect at lower concentrations. In addition, both honeys increased intracellular reactive oxygen species generation. The results of this last study suggest a potential chemopreventive action of bitter honey, which could be useful in the perspective to develop chemopreventive agents for colon cancer.

Further results by the same authors [62,63,64] on the phytochemical composition and anticancer effects of bitter honey on cellular proliferation, cell cycle, and apoptosis in human colon adenocarcinoma and metastatic cancer cells showed that kaempferol and gallic acid were the major phenolic compounds. Bitter honey evidenced higher cytotoxic and anti-colonogenic effects in a time- and dose-dependent manner, showing a chemo-preventive action on different colon cancer cell models, once again demonstrating its potential use in cancer prevention.

Antioxidant activity was also proved for strawberry tree honey from South Portugal. In this report, the bioactive compounds, evaluated as total phenolic (94.47 mg gallic acid/100 g) and total flavonoid content (5.33 mg quercetin/100 g), showed a radical scavenging activity (DPPH assay) of 43.46% and an antioxidant activity of 18.85 mg ascorbic acid equivalent/100 g and 9.92 mg quercetin equivalent/100 g [65].

A strong positive relationship was found between total phenolic content (TPC) and 1,1-diphenyl-2-picrylhydrazyl (DPPH), TPC and concentration of homogentisic acid homogentisic acid (HGA), and between DPPH and HGA in strawberry tree honey from Croatia [66,67]. A further study on Croatian strawberry tree honey revealed fifty-two phenolic compounds (twenty-seven phenolic acids and twenty-five flavonoids). The overall results evidenced levels of total phenolic content of 1038 mg gallic acid equivalents per kg of honey and a radical scavenging activity of 3.32 mmol Trolox equivalents per kg of honey [56].

Based on literature, it is evident that the antioxidants properties of Sardinian bitter honey are due mostly to the presence of homongentisic acid in its composition. This acid was also found in many unifloral and multifloral Turkish honeys, but the amounts were generally very low, except in the case of thyme honey, which showed levels of 384.06 ± 2.10 mg/kg comparable to those of strawberry tree honey. Among strawberry tree honeys, Sardinian bitter honey showed higher content of this phenolic acid, ranged between 377.6 ± 20.1 and 520.3 ± 17.2 mg/kg [58], compared to 310.9 ± 85.6 mg/kg for Croatian ones [67]. Although homogentisic acid represents the main phenolic compound in Sardinian bitter honey, we cannot exclude a synergistic antioxidant effect with other phenolic compounds present in this honey.

## 6. Conclusions

Sardinian bitter honey from strawberry tree (*Arbutus unedo*) nectar is a source of bioactive substances that showed interesting in vitro and in vivo biological properties. This honey, according to literature, is effective in contrasting microbial infections and potentially to hinder malignant cellular proliferation through the modulation of several molecular pathways, and also to counteract several risk factors of cardiovascular disease. This honey, thorough its main antioxidant compound, may have a great potential as functional food and for its promising pharmacological uses aimed to preventing various important disorders or diseases and aging. Regardless, an in-depth understanding of the factors and the mechanisms that determine the effects of this honey will be of crucial importance to promote its use for healing purposes by the consumers in order to prevent diverse important disorders or diseases.

In Sardinia, the shrublands are characterized by strawberry tree presence cover a surface of about 450,000 hectares; the potential bitter honey yield is therefore very high despite the seasonal limits of production. However, until now, no regional action was promoted for the marketing valorization of this honey. The tools to promote it properly would be there, e.g., the Protected Designation of Origin (PDO) represents a very important EU certification recognized for products originating in a specific geographical zone. We might add that for this Sardinian honey, given its long-standing historical reputation and other characteristics, such as a well-defined pollen spectrum for Sardinia, attributing it to its geographical origin should not be difficult. Moreover, the most recent findings on its therapeutic properties should encourage producers to undertake efforts to promote it in order to protect and favor a productive system and the economy of the territory, as has been done for other important honeys in the world. The most important example is probably the manuka (*Leptospermum scoparium*) honey from New Zealand, that has been extensively studied for antibacterial and antioxidant activity, as well as for the therapeutic properties, i.e., wound healing mechanisms. Manuka honey as the bitter honey contains numerous phenolic compounds, flavonoids, and phenolic acids, and other compounds such as methylglyoxal [68], which induces non-peroxide antibacterial activity even at very low concentrations [69,70]. Methylglyoxal is the active ingredient measured in manuka honey for marketing, the dosage of which determines its selling price. Similarly, homogentisic acid for strawberry tree honey could be the bioactive substance through which the pharmaceutical use of this honey could be promoted.

## Figures and Tables

**Figure 1 antioxidants-10-00506-f001:**
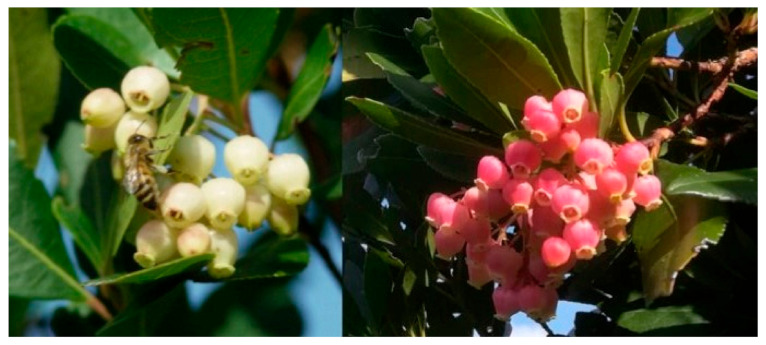
Strawberry tree (*Arbutus unedo* L.) flowers of two distinct varieties from Sardinia (Photos: Alessandro Lampis).

**Figure 2 antioxidants-10-00506-f002:**
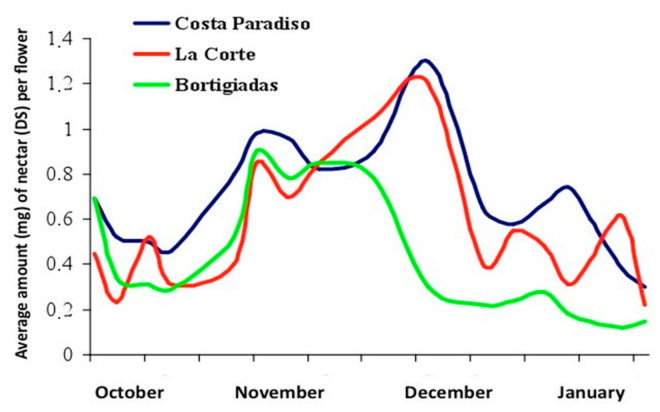
Trends in nectar secretion of strawberry tree (*Arbutus unedo* L.) flower in different locations of northern Sardinia [9].

**Figure 3 antioxidants-10-00506-f003:**
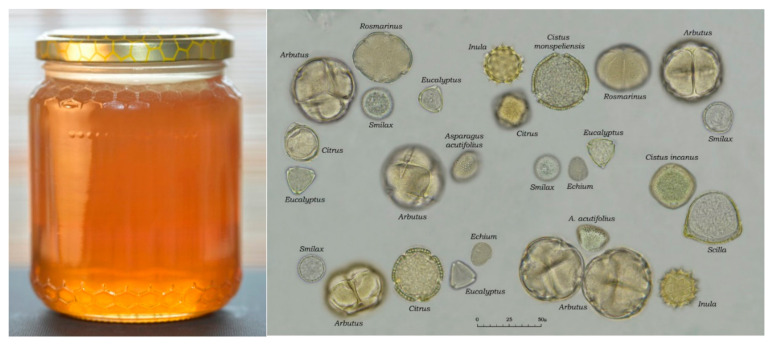
A jar of Sardinian bitter honey and its common melissopalynological spectrum (Photos: Nicola Palmieri).

**Figure 4 antioxidants-10-00506-f004:**
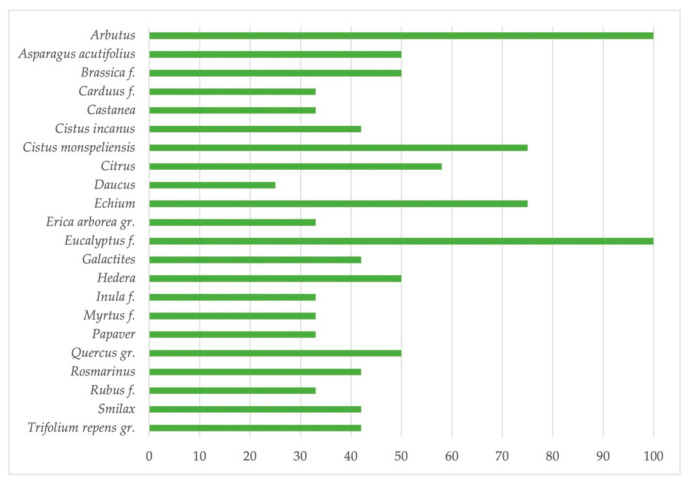
The most frequent pollens recorded in Sardinian bitter honey samples (expressed as % of honey samples) [14].

**Figure 5 antioxidants-10-00506-f005:**
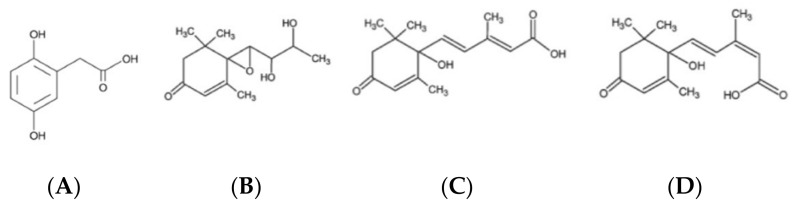
Structures of floral markers of Sardinian bitter honey: (**A**) homogentisic acid (2,5-Dihydroxyphenylacetic acid) (**B**) unedone, (**C**) (±)-2-*cis*,4-*trans*-abscisic acid, (**D**) (±)-2-*trans*,4-*trans*-abscisic acid) [16].

**Figure 6 antioxidants-10-00506-f006:**
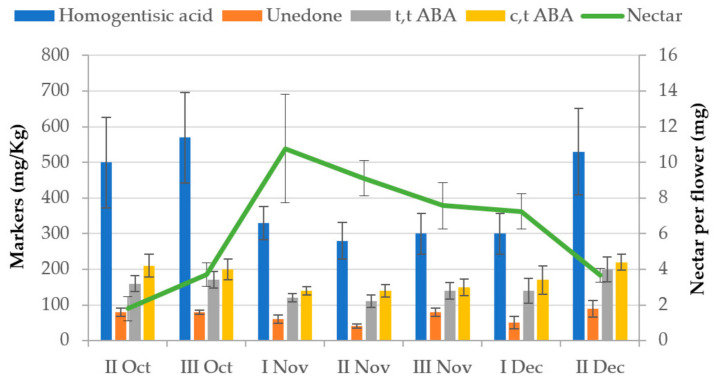
Concentration (mg/kg ± SE) of different markers (HGA—Homogentisic Acid; Unedone; t,t ABA—(±)-2-trans,4-trans-abscisic acid, t,t ABA; c,t ABA—(±)-2-cis,4-trans-abscisic acid); in nectar of strawberry tree (*Arbutus unedo* L.), flowers, and trends (means among different stations ± SE) of nectar secretion during the flowering period (decades) [18].

**Figure 7 antioxidants-10-00506-f007:**
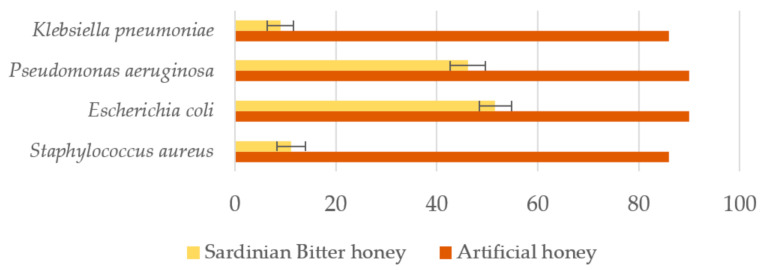
Comparison of growth levels (colony-forming unit) [46] of the different bacterial cultures by visual assessment of partial inhibition on the agar plate with the concentration of honey [44,47] between Sardinian bitter honey from strawberry tree (*Arbutus unedo* L.) and artificial honey (sugar solution) [45].

**Figure 8 antioxidants-10-00506-f008:**
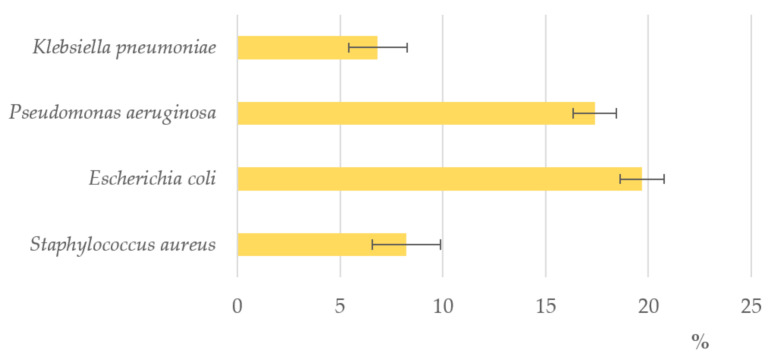
MIC—Minimum Inhibitory Concentration (% *v*/*v*) of strawberry tree (*Arbutus unedo* L.) honey (STH) from Sardinia (bitter honey) against different bacterial cultures [42,43].

## Data Availability

The data presented in this study are available on request from the corresponding author.

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
