# Peer review of "The Sardinian Bitter Honey: From Ancient Healing Use to Recent Findings"

_antioxidants, 2021, doi:10.3390/antiox10040506_

Round 1

Reviewer 1 Report

The manuscript is well written, gives an overview of current knowledge on Sardinian bitter honey and should be of great interest to the readers.

A legend in figure 6 should be checked. In this version of the manuscript, a fourth series (yellow) is not visible.

Author Response

Dear Reviewer 1,

We appreciate that the reviewer considers our topic as being very interesting.

Now the fourth series (yellow) is visible. The colour of fourth series was changed to make it more visible.

Best Regards

Ignazio Floris

Reviewer 2 Report

The Manuscript submitted by Floris et al is a review (more like mini-review in fact) about the properties of Sardinian bitter honey in comparison with other types of honey. The bitter honey is a relatively famous type of honey, but which has not been investigated well enough, although it was shown to have high biological activities (antibacterial, antioxidant). The authors try to promote this honey and the importance to valorise it more. 

The work is rather an average contribution to the field and I am not 100% convinced that it meets the standards of the journal Antioxidants, but I leave the final decision of suitability to be taken by the Editor. I decided to recommend major revision for improvement, because the paper has some potential. If not for Antioxidants, maybe for other Journal. Please see attached the PDF with the comments.

Mainly, tables are missing from the paper and are necessary, the antibacterial chapter is not very clear and I think it has some mistakes. Some parts are not clear, if the bitter honey is better or worse compared to which honey. Here the tables would be helpful, as mentioned.

Also, considering the Journal, the authors should have focused more on the antioxidant characteristics of the honey and maybe even describe it before the antibacterial one. There is much less information presented about the prooxidant effects of polyphenols which is also a factor known to contribute to antimicrobial effects in honey.

The authors should describe also other antioxidant methods that have been used, such as FRAP (I gave one reference in the text) and discuss the results by comparing also the methods and the biological significance of the methods. This would be more relevant for the Journal.

There are also some minor issues mentioned in the PDF and the authors should recheck the English because there are some flaws.

Author Response

Dear Reviewer 2,

We thank the reviewer who correctly captured the target and the relevance of the review.

Please, attached you will find, point by point, our responses to the specific comments.

Best Regards

Ignazio Floris

Reviewer 3 Report

Dear authors,

the review describes the new findings upon the bitter honey produced at the Sardinian island. The paper is well structure and organized, and the written english is appropriate. 

Clearly the authors have a huge background on the study of the aforementioned topic. Morevover, the bibliographic information applied in the manuscrit support all the discussion through the biological properties of the studied product.

Author Response

Dear Reviewer 3,

We thank the reviewer for the strongly positive evaluation of our manuscript.

Best Regards

Ignazio Floris

Round 2

Reviewer 2 Report

The authors managed to improve their manuscript and answered most of the observations.

I have one minor comment with respect to the title. The definite article "the" is not necessary in the context provided, at least the second one: "[...] from (the) ancient healing use to recent findings" should be used in my opinion.

Author Response

Thank you very much for the last useful suggestions.

The title has now been changed as follows: The Sardinian bitter honey: from ancient healing use to recent findings.

Best Regards

Ignazio Floris